# Superblock Design and Evaluation by a Microscopic Door-to-Door Simulation Approach

Ngoc An Nguyen [iD], Joerg Schweizer [iD], Federico Rupi *[iD], Sofia Palese and Leonardo Posati

Department of Civil, Chemical, Environmental and Materials Engineering, University of Bologna, 40126 Bologna, Italy; ngocan.nguyen2@unibo.it (N.A.N.); joerg.schweizer@unibo.it (J.S.); sofia.palese2@unibo.it (S.P.); leonardo.posati@studio.unibo.it (L.P.)
* Correspondence: federico.rupi@unibo.it

**Abstract:** The present study contributes to narrowing down the research gap in modeling individual door-to-door trips in a superblock scenario and in evaluating the respective impacts in terms of travel times, modal shifts, traffic performance, and environmental benefits. The methods used are a multiple-criteria approach to identify the superblocks and a large-scale, multi-model, activity-based microscopic simulation. These methods were applied to the city of Bologna, Italy, where 49 feasible superblocks were identified. A previous large-scale microscopic traffic model of Bologna is leveraged to build a baseline scenario. A superblock scenario is then created to model five proposed traffic intervention measures. Several mobility benefit indicators at both citywide and superblock levels are compared. The simulation results indicate a significant increase in walking time for car drivers, while the average waiting time of bus users decreases due to the increased frequency of bus services. This leads to a noticeable car-to-bus shift. In addition, absolute traffic volumes and traffic-related emissions decreased significantly. Surprisingly, traffic volumes on the roads around the superblocks did not increase as expected. In general, this research provides scientists and urban and transport planners with insights into how changes in door-to-door travel times of multi-modal trips can impact individual travel behavior and traffic performance at a citywide level. However, the study still has limitations in modeling the long-term effects regarding changing activity locations within the superblocks.

**Keywords:** large-scale microscopic traffic simulation; superblock; car-free zone; low emission zone; activity-based model (ABM); door-to-door travel time

## 1. Introduction

### 1.1. Introduction to Superblocks and Their Analysis

Cities are home to over half of the world's population, generate around 70% of global carbon dioxide ($CO_2$) emissions [1], and the transport sector contributes 51% of the $CO_2$ emission in major cities [2]. It has been acknowledged that the "Avoid–Shift–Improve" approach is strategic to achieve net-zero carbon by 2050 under the 2015 Paris Agreement [3–5]. This holistic approach is focused on (i) "avoiding" unnecessary travel needs through urban redesign, (ii) "shifting" to sustainable transport modes, and (iii) "improving" the travel experience of sustainable modes [5–8]. The superblock model integrates "avoid" and "shift" strategies by reclaiming public space and promoting sustainable mobility [9]. In addition, it follows the "improve" strategy by enhancing public transport (PT) supply and services [10].

Superblocks transform grid urban road networks into pedestrian-friendly neighborhoods, with local roads reserved for active mobility and arterial roads handling motorized and cross-through traffic, as illustrated in Figure 1. Superblocks promote active mobility by placing parking outside the blocks, permitting access for bicycles, taxis, shared electric cars, and freight deliveries, while private car access is limited to residents only. The implementation of superblocks can reshape citywide traffic flow and modal split. However,

without changes in modal choices, traffic volumes on bordering arterial roads are likely to increase [11]. For this reason, PT services need to be improved or new alternative modes need to be introduced. In this way, the total carrying capacity of the arterials increases and alleviates congestion [12].

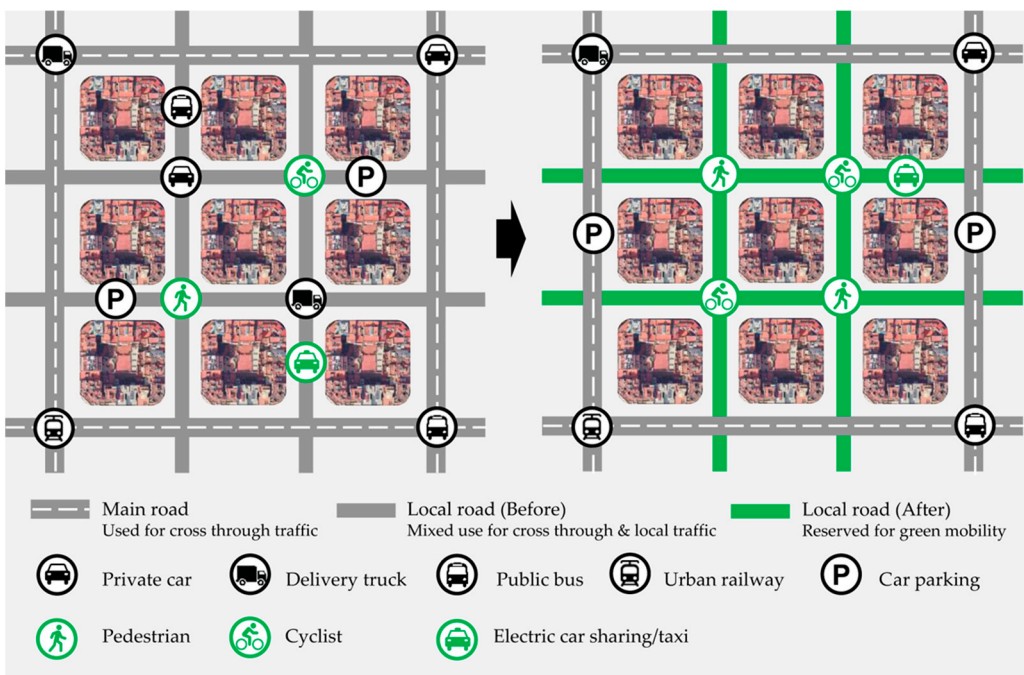

**Figure 1.** The reorganization of the street grid when implementing the superblock concept.

Superblock implementation, only at the city scale, is expected to have significant global effects [13]. In addition, traffic rerouting alters the travel demand and individual movement [14]. Therefore, the transport model used to analyze the impacts should (i) cover the entire urban area to capture large-scale changes, (ii) be sensitive to changes in the local environment of individuals, and (iii) include active modes and multi-modal trips to model trip chains within or between superblocks.

In recent years, large-scale microsimulation models, sometimes termed "transport digital twins", have been developed, which can reproduce multi-modal trips of individuals, such as the scenarios of Bologna, Cologne, and Luxembourg [15–17]. These models would be able to adequately represent changes in the transport supply in a what-if scenario, as well as simulate and quantify their impacts. However, multi-modal, large-scale micro-simulation scenarios are complex as they are composed of many details and necessitate the merging of different datasets from different sources. The challenges in building, calibrating, and validating such scenarios are reported in [18].

A baseline scenario of the city of Bologna, based on the micro-simulation environment SUMO [19], has been developed, improved, and recalibrated in [15,20,21]. SUMO version 1.12 is an open-source microsimulation software maintained by German Aerospace Center—DLR in Berlin, Germany. A what-if scenario with superblocks that cover a substantial part of Bologna has been created with established criteria. This approach enables the simulation of travel experiences, the estimation of mode choices, and the door-to-door travel time of individuals. With these results, the mobility benefits and impacts on the performance of the entire transport system can be evaluated.

### 1.2. State of the Art

The superblock model is not entirely new, but evolving from varied approaches, all aiming for "human urbanism". The "local network" scheme proposed in the 1960s by Radburn in Buchanan's book *Traffic in Towns* [22] can be considered a precursor to superblocks.

The scheme separated vehicles and pedestrian circulations into mixed-use zones, where all destinations are accessible without crossing roads. It was only decades later that various implementations of this idea emerged, such as "living streets", "shared space", or "city quarters" with prohibited car circulation [10]. Various versions of these schemes have been found in numerous European cities, such as "share-spaces", or "car-free development" in the United Kingdom [23,24], "woonerf" in the Netherlands [25], "car-free or car-reduced housing areas" in the Freiburg-Vauban, "Kiez blocks" in Germany [26,27] and Vienna [28], and "micro-districts" in Russia [29]. This concept has also been explored in various urban contexts, such as in China (Asia) [30], Abu Dhabi (Middle East) [31], Panama [32], and the United States of America [33]. However, the superblocks being deployed in Barcelona, Spain, are the largest in size, covering the entire city core (100 km$^2$), with 503 superblocks planned and nine implemented or in progress [10,34,35]. The Barcelona scheme is detailed further as the most studied case, inspiring the present work as well.

The Barcelona superblocks are configured by nine-block grids (3 × 3 blocks, as shown in Figure 1), with approximately 400 m side length. However, this may vary in peri-urban areas and typically depends on the structure and layout of the city [33]. The diversity of land uses, services, and population density guide superblock creation [36]. A sustainable superblock is characterized by a high population density (10,000 inhabitants per km$^2$) and a large building footprint (>30%) [37,38]. In fact, a Barcelona superblock, home to over 6000 inhabitants, enhances accessibility due to seamlessly integrated public services [39].

Several traffic calming measures are introduced within superblocks, including (i) restricting motorized access and cut-through traffic, (ii) prohibition of on-street parking, (iii) traffic calming within superblocks and adjacent areas, and (iv) lowering travel speed down to 10–20 km/h on local roads [11,39]. The inner roads remain accessible to residential traffic, as well as service, emergency, and loading/unloading vehicles under specific conditions [10]. The restrictions facilitate easier non-motorized access, and, in the long term, advanced transport modes are suggested for improved mobility within the block [12,40].

Additional measures should be taken to accommodate the increase in traffic volume on the arterial roads, such as (i) provision of segregated and prioritized lanes for high-capacity PT lines; (ii) increasing the frequency of bus services to reduce waiting time and increase capacity; (iii) segregating infrastructures for cyclists and pedestrians to promote the safety of active modes [41]; (iv) providing multi-level parking spaces to facilitate seamless intermodal transfer at transit stations bordering superblocks; and (v) relocating bus stops near the main intersections of superblocks to increase accessibility for the citizens [9,39].

Researchers have used various methods to evaluate the mobility impacts of the superblock model. A four-step modeling approach with the macroscopic traffic modeling software TransCAD [42] has been used to model 503 superblocks in Barcelona [43]. A "desired" or expected reduction in the car mode share by 21% and a significant increase in walking (10%), cycling (67%), and PT (3.5%) has been assumed [39]. The desired modal shares have not been verified, as only 6/503 superblocks have been implemented. In any case, the adopted policy changes and concrete measures have not been captured by the macroscopic modeling, which has not been able to quantify the modal shift.

Mueller et al. [9] conducted a quantitative health impact assessment and evaluated baseline and superblock scenarios based on a hypothesized modal shift. Their study assumed a drop of 19.2% in private traffic volume, for substitution, an increase in PT by 1.8%, cycling by 0.4%, and walking by 2.8% after the introduction of the 503 superblocks in Barcelona. The model predicted 667 fewer annual deaths (291 from NO$_2$, 163 from noise, and 117 from heat exposure). It also estimated that 36 annual deaths could be prevented if 65,000 persons shifted from private cars to PT and active modes.

Rodriguez-Rey et al. [11] assessed the impacts of Barcelona's transport planning strategies, including the introduction of superblocks by using coupled macroscopic traffic and emission models. The study used a trip-based model [44] and the macroscopic traffic simulator VISUM [45] assumed reductions of 0% and 25% in travel demand after the introduction of superblocks. The study concluded that, without a traffic demand reduc-

tion, the overall traffic emissions in the city remained unchanged. While $NO_x$ emissions were reduced within the superblock, they increased by 17% on the main roads due to traffic diversion and new bottlenecks. In contrast, significant emission reductions would require both a drop of 20% in traffic demand and optimistic fleet renewal, coupled with all associated measures.

Studies on the impacts of superblock interventions on climate and health benefits have also been conducted in the city of Vienna, Austria, where the superblock concept is being expanded [46]. Traffic modeling has been used to estimate how superblocks could affect the mobility behavior and travel distance of citizens in three potential sites. A latent class model [47] has been adopted based on diary survey data [48]. The mode-choice model has predicted the behavior of people circulating in/out of the superblock areas with a reduction of 2–5% in traveled car kilometers and model shift to active modes.

In a recent effort to improve the transport models of superblocks, a mesoscopic agent-based model using the MATSIM software has been employed to assess how sequentially implementing 46 superblocks would impact the mode choices consisting of cars, pedestrians, bicycles, and PT in the city of Vienna [49]. By simulating 12.5% of the city population using a linear polynomial model, the study predicted a linear decrease in car trips of 100 trips/day within the superblock, with car users largely shifting to walking and PT. However, due to a lack of validation, the study called for further research efforts in evaluating traffic performance on arterial roads.

The above-cited research is summarized in Table 1, which reveals the following research gaps: (i) The details of the superblock implementation, in particular the single stages (e.g., walking, waiting, riding) of multi-modal door-to-door trips in a superblock area, have been insufficiently modeled. This has led to difficulties in the explanation of individual travel behavior and mode choice. Some recent mesoscopic models of Vienna have narrowed down this gap, but detailed and large-scale models have not yet been validated; (ii) The simulation of door-to-door travel times of a large-scale network is still missing, which is needed to quantify the experience of individuals and vehicle interactions to assess the traffic flow and link travel times.

**Table 1.** Comparison of published traffic simulations to evaluate the impacts of the superblock model.

| # | Case Study | Superblock Scale | Traffic Model | Superblock Mode-Share Assumptions or Model | Traffic Simulator | Reference |
|---|---|---|---|---|---|---|
| 1 | Barcelona, Spain | 503 superblocks | Four-step model, O-D matrix | "Desired" mobility model | TRANSCAD, macroscopic model | [43] |
| 2 | Barcelona, Spain | 503 superblocks | Not applicable | Hypothesized modal shift | Not applicable | [9] |
| 3 | Barcelona, Spain | 6 superblocks | Four-step model, 2017 O-D matrix from mobile phone | Assumed reduction of 0% and 25% circulating vehicles | VISUM, macroscopic model | [11] |
| 4 | Vienna, Austria | 3 superblocks | Latent class model | Assumed reduction in kilometers traveled by car and shift to active modes | Not applicable | [46] |
| 5 | Vienna, Austria | 46 superblocks | Agent-based model, 12.5% population | Based on a linear polynomial model to fit curves for each mode | MATSim, mesoscopic model | [49] |
| 6 | Bologna, Italy | 49 superblocks | Activity base, disaggregation of O-D matrix, GTFS, 100% population | Mode-share model based on utility function including individual door-to-door trip time | SUMOPy/SUMO, large-scale microscopic model | This paper |

To fill this research gap, the aim of this paper is to present a multicriteria superblock design and a microscopic superblock model with the following characteristics: (i) all essential elements of a superblock design are represented in the model; (ii) the mode share can be determined as the sum of individual mode-choice decisions based on utility function;

(iii) door-to-door trip-times of all individuals can be simulated; (iv) congestion analysis and a more precise impact assessment based on average flow, and realistic speed profiles of individual vehicles can be simulated at an urban scale. The proposed microscopic activity-based model works at a disaggregated person level rather than at an aggregate zone level, like most trip-based models [50]. Therefore, it is highly sensitive to changes in traffic supply that impact daily travel behavior [51]. It has been acknowledged in the literature that the microscopic simulation approach can address the current limitations in properly estimating changes in model splits, travel demand, and traffic flow in superblock scenarios [11,52].

The proposed modeling methods include superblock identification, infrastructure modeling, demand modeling, modal split modeling, and subsequent analysis of baseline and superblock scenarios. The modeling is demonstrated on a large-scale microsimulation model of Bologna [15], which is leveraged with significant improvements to build the baseline scenario. However, the modeling framework is flexible and can be readily adapted to different urban settings.

The remaining parts of this paper are structured as follows: Section 2 describes the methods in modeling door-to-door travel time and mode-share estimation in the microscopic traffic model, proposes an approach to identify the superblock configuration as well as the baseline model of Bologna, its calibration, validation, and application to create and model the superblocks. Section 3 presents the simulation results of the baseline scenario versus the superblock scenario, focusing on the model splits, changes in door-to-door travel times, evaluations of traffic performance, and traffic-related air emissions, followed by a discussion. Finally, some conclusions are drawn in Section 4.

## 2. Materials and Methods

This section explains the microscopic traffic simulation model and the multicriteria method used to create the superblocks. Both methods are applied to the metropolitan area of Bologna. Thereafter, the validation of the baseline scenario and the integration of the traffic intervention measures into the superblock scenario are described.

### 2.1. Microscopic Door-to-Door Simulation Approach

The study uses the microscopic simulation suite SUMOPy/SUMO, which is able to simulate every single vehicle and person, replicating a city's real population statistically. SUMOPy is a simulation tool and part of the open-source software SUMO. The baseline scenario is an enhancement and upgrade of a previously developed activity-based model [15]. It incorporates a demand model generating the plans of each synthetic person with detail on how to travel from door to door and a supply model encompassing all road network details derived from Openstreetmap (OSM) data [53], representing all road links, nodes, lanes and associated attributes, configurable traffic lights, parking, and PT imported from GTFS.

This paper defines "door-to-door" travel time for different vehicle users by considering the entire journey from origin to destination, as shown in Figure 2. For bus trips, this is the most complicated journey, including (i) the walking time from his/her origin (e.g., home or workplace) to the nearest bus stop, the so-called "access time"; (ii) the waiting time at the first bus stop, and at the transfer stops (if any); (iii) the riding time on the bus to a transfer point (if any) or to the final destination; and (iv) the walking time from the last bus stop, the so-called "egress time" to his/her final destination. For car trips, this includes three stages: (i) the walking time from his/her origin to the parking lot; (ii) the driving time from the parking lot to places near his/her destination where he/she can leave the car; and (iii) the walking time from the parking lot to the destination. Both car and bus door-to-door travel time approaches are similar to those in [54]. For pedestrian, bicycle, and motorcycle trips, the door-to-door travel times are simpler, including walking time (for pedestrians) or riding time (for cyclists and motorcyclists) from origin to destination. The total door-to-door travel time of a single person can be broken down into the following components:

$$T = t_{walk,s} + t_{wait,s} + t_{ride,s} \tag{1}$$

where $T$ is the total door-to-door travel time of a single person; $s$ represents the transport modes in the microscopic simulation model; $t_{walk,s}$ is the walking time of a person using mode $s$; $t_{wait,s}$ is the waiting time of a person using mode $s$, which is applied for bus users only; and $t_{ride,s}$ is riding time of a person using mode $s$.

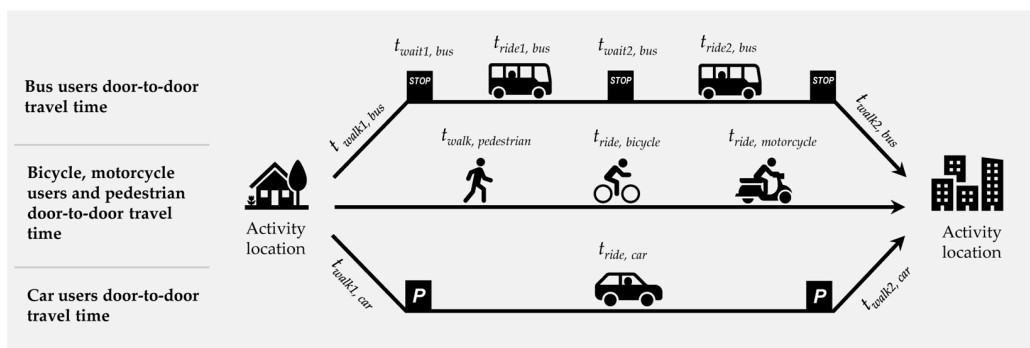

**Figure 2.** The concept of the door-to-door travel time in the microscopic simulation model of different vehicle users. An example of a single trip.

The micro-simulation method and the mode-share estimation are a core part of the study. The traffic micro-simulation and the mode choice are interdependent as the mode choice will impact traffic flows and vice versa. The methods employed in this study are basically the ones described in [15], with some important alterations concerning the micro-simulation method.

The micro-simulation tracks individual movements and vehicle interactions from origin to destination. The vehicle routes should be determined such that the vehicle flows on each edge to become realistic. The conventional method used to determine realistic flows is called the "user equilibrium" [55]. With micro-simulations, the determination of the routes leading to a user equilibrium requires running an iterative process [56]. However, as this is a slow and computationally expensive process, an en route routing process was adopted, where vehicles can change routes during the simulation, dependent on the current traffic situation. Before the simulation, the route of each vehicle and person is stochastically pre-determined—shortest travel time assignment with some randomness in the free-flow edge travel times; during the simulation, and in certain intervals, a certain share of the vehicles try to find a faster route based on the current and alternative edge travel times. In the end, the overall traffic distribution seeks user-equilibrium-like traffic patterns. This simulation method allows the achievement of realistic traffic flows with a single simulation run, which is essential to the feasibility of the presented method. This is because the determination of a realistic mode choice requires that all persons "experience" the door-to-door travel time of their mobility plans. This means a particular scenario must be run several times, while persons may change mobility plans until at least a significant part of the population has executed all her/his feasible mobility plans.

The mode-choice model used in this study aims at realistically forecasting mode shares if door-to-door times change. The model assumes that all conditions of a person are maintained across scenarios (keeping the same activity locations and all other socio-economic attributes), with only travel times varying by mode. Thus, a maximum utility model has been adopted where each mode has a utility composed of a mode-specific attribute $\alpha_s$ and a time-dependent component, as recapped below.

$$U_{s,i} = \alpha_s - \beta \cdot T_{s,i} \tag{2}$$

where $U_{s,i}$ is the utility function of plan $s$ for person $i$; $T_{s,i}$ is the plan execution time (e.g., the simulated door-to-door time) of plan $s$ of person $i$; and $\beta$ is the value of time (VoT) for all people and plans.

The $\alpha_s$ parameters were calibrated such that the population in the baseline scenario (after simulation of all travel plans) reproduces the latest city's mode-share estimates. The calibrated model is then used to estimate the mode share in the superblock scenario.

## 2.2. Superblock Configuration Identification

A multiple-criteria approach, which is widely used in [33,37,38,49], was adopted to identify potential superblocks in this paper, as indicated in Figure 3. First, the most suitable superblock size of 400 m × 400 m was preferred to formulate a list of initial candidates. However, the dimension of potential superblocks may reach 700–800 m due to urban design constraints formed by asymmetrical road shapes. The objective is to ensure that PT stops placed along the main road are highly accessible within an 8 min walk and at a comfortable walking speed of 3 km/h. Next, four critical sustainability criteria were assessed to optimize potential superblock boundaries: (i) road network hierarchy; (ii) population density; (iii) building footprint coverage; and (iv) current PT network.

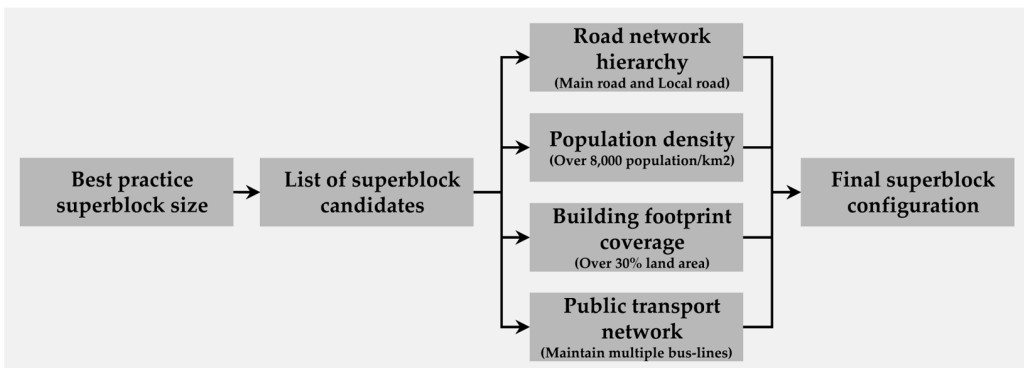

**Figure 3.** Proposed approach in selecting the superblock configuration in this paper.

To build a 2-level road hierarchy of main and local roads, the OpenStreetMap (OSM) [53]—an open-source project that contains most of the transport network attributes [15] —was used, in particular its "highway" element. First, all highways with a footpath attribute were filtered. The roads labeled as "motorway", "trunk", "primary", "secondary", or "tertiary" were considered as the main road class, while the remaining roads belonged to the local road class. Then, Criterion 1 could be applied: superblocks must be surrounded but not crossed by main roads to avoid through traffic and maintain traffic circulation around the blocks.

Superblocks are typically proposed for high-density neighborhoods [38]. The population density available in the census database was determined. Then, Criterion 2 could be applied by overlaying the road network hierarchy layers to identify superblock candidates that meet the population density threshold. Next, the buildings and land use data obtained from OSM integrated into the scenario of a previous study [15] allowed us to measure the building footprint coverage of superblock candidates. Criterion 3 could be applied by selecting superblocks with a building coverage greater than the threshold of 30%, as recommended for sustainable superblocks.

Finally, PT networks were overlaid onto the first three layers. Applying Criterion 4, the boundaries of the final superblocks were adjusted to ensure the maintenance of the existing bus routes and minimize service disruptions. All roads served by multiple bus lines are maintained and associated with superblock boundaries, while single bus routes crossing the superblock will deviate to follow main roads.

### 2.3. Bologna Study Area and Proposed Superblock Configuration

The superblock modeling is applied to Bologna—a medium-sized Italian city. The wider metropolitan area covers 3703 km$^2$ and has a population of about 1.02 million inhabitants [57,58] Meanwhile, the municipality of Bologna itself covers a total area of 141 km$^2$ and had 390 thousand inhabitants and a population density of 2769 inhabitants/km$^2$ in 2022. The historic center, inside the inner ring road, has a population density of 20,424 inhabitants/km$^2$ [59]. The road network of the city of Bologna is denser within the outer ring road, encompassing motorways, major federal roads, provincial roads, and urban roads. The latest city mode share is as follows: walking, 21.30%; bicycle, 6.90%; PT, 25.60%; car, 35.60%; and motorbike, 10.60% [57]. The city government is introducing low-emission zones and planning 30 km/h speed limits in the urban core to increase road safety and promote active mobility [60].

The four criteria for selecting superblock configuration, proposed in Section 2.2, were applied to the Bologna case study. Following the first criterion, 14,246 main roads and 13,165 local roads were classified, as shown in Figure 4a. Then, by using the 2022 Bologna population data covering 90 statistical zones [59], a population density threshold of greater than 8000 people/km$^2$ was applied for Criterion 2, which is slightly lower than the threshold of 10,000 people/km$^2$ proposed in [38], to reflect the lower population density in Bologna city, as shown in Figure 4b. Building coverage greater than 30% was estimated and applied for Criterion 3, as shown in Figure 4c. Finally, the 2023 Bologna bus network extracted from the GTFS data obtained from the Bologna PT operator (Tper) [61] was used to overlay the first three layers.

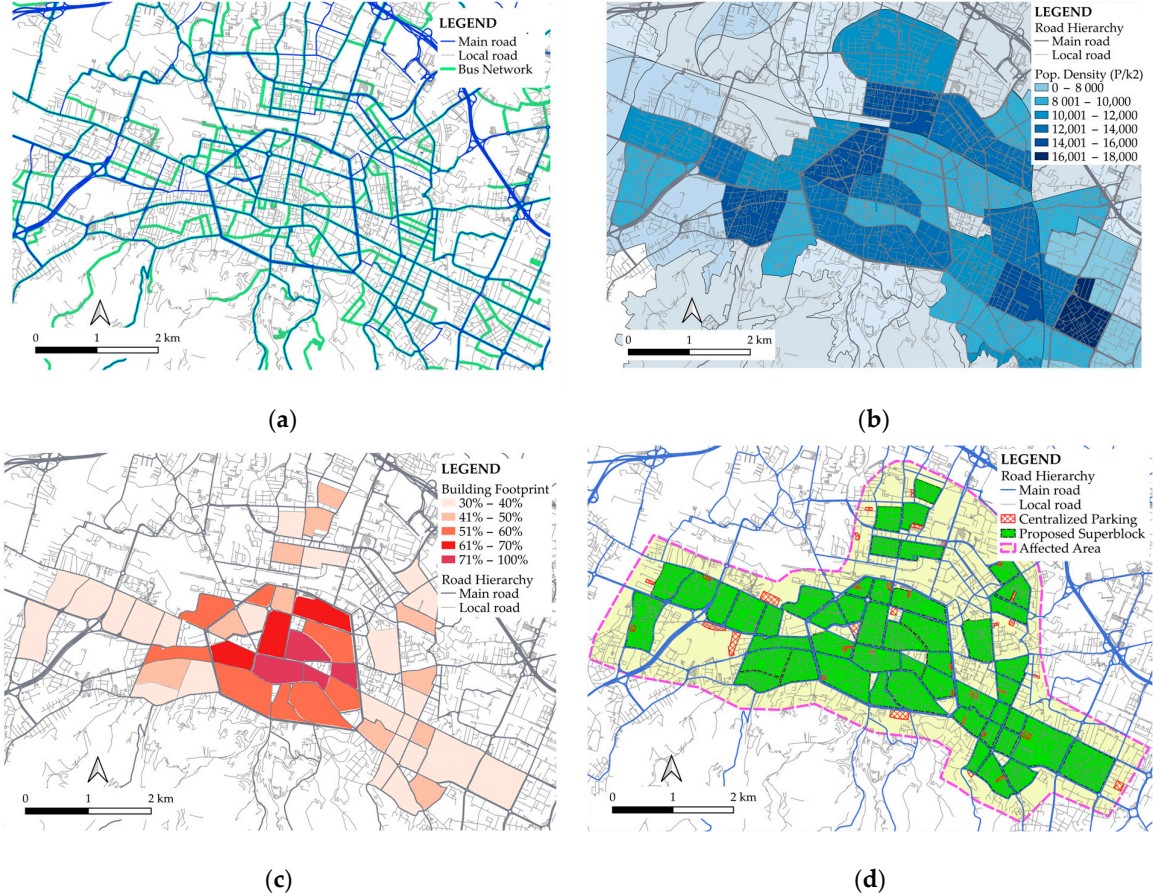

(a)　　　　　　　　　　　　　　(b)

(c)　　　　　　　　　　　　　　(d)

**Figure 4.** (**a**) The hierarchy of the road network in Bologna (main roads—blue color, local roads—gray color) and urban bus network (green color). (**b**) The population density of Bologna in 2022. (**c**) The building footprint ratio of superblock candidates. (**d**) Proposed configuration of 49 superblocks in Bologna (green area), affected areas (yellow area), and proposed centralized parking (gridded red area).

As a result, a total of 49 feasible superblocks were identified across Bologna city, covering a total area of 10.7 km², corresponding to an average superblock area of 0.22 km², as indicated in Figure 4d. All superblocks are surrounded by main roads for cross-through traffic and are served by frequent bus services. The local roads inside the superblocks are reserved for community activities and active mobility. Within the inner ring roads, the historic center, 16 superblocks were identified. The superblocks are evenly distributed in a west–east direction, with seven superblocks scattered to the north of the city center. The implementation of the superblocks is explained in Section 2.5.

### 2.4. The Baseline Scenario and Its Validation

With respect to the model in [15], important changes and modifications were carried out to improve the accuracy of the model: (i) extensions—the road network model was extended to surrounding urban areas of Bologna; (ii) refinements—the road network converted from OSM was further refined with SUMO's "netedit" [62] to capture most up-to-date road network configurations, focusing on access rights, lane connectivity, zebra crossings, and traffic light programs; (iii) on-street parking adjustments to facilitate the plan generation of car users.

As a result, the Bologna supply model comprises a road network characterized by 32,409 road links, 14,724 intersections corresponding to 530 signalized junctions, and 292,944 on-street parking spaces. The entire PT network of 234 bus lines was modeled based on data from GTFS provided by the local PT operator Tper [61].

The transport demand was generated for the morning from 6:00 a.m. to 8:00 a.m. by disaggregating the OD matrices of cars, motorcycles, buses, bikes, and pedestrians. The raw OD matrices obtained from the 14th population census [63] were scaled to the year 2018 by considering annual population growth rates in the study area. The demand model consists of the internal demand and the external demand. The internal demand is represented by all trips within the urban area of Bologna city made by the synthetic population. The internal demand comprising 167,062 people was generated by disaggregating the OD matrices. One or several plans for a single person were then generated. Each citizen has at least one plan, called the "preferred plan", using a transport mode prescribed by the mode of the OD matrix. However, it is also possible to generate plans with other feasible modes if the person possesses the required vehicle—walking and bus are feasible for all. As a result, 448,597 plans were generated for the 167,062 people, consisting of 19.29% car plans, 25.55% bus plans, 11.73% bicycle plans, 6.18% motorcycle plans, and 37.24% walking plans. It is noted that plans are door-to-door and can contain multiple modes; for example, a bus or car trip includes walking to the stop or parking, respectively.

The external demand was created with 32,193 car and 2261 motorcycle trips between the urban area and the extra-urban areas as well as between extra-urban areas that pass through the urban area. These trips are generated by a simple disaggregation of the OD matrix of the respective mode, where the distribution over the edges within the zone of origin and destination depends on the length of the edges.

As mentioned in Section 2.1, the plan choice model in Equation (2) was calibrated to guarantee that the population's plan choices reflect the latest mode shares in the city as reported in [57]. The value of time was determined, as shown in Table 2. Therefore, the $\beta$ value of 0.002 EUR/s was adopted based on the relevant Italian studies [64,65].

**Table 2.** Recalibrated parameters of the utility function of the mode-share model.

| $\alpha$1 Car (ref.) | $\alpha$2 Bike in EUR | $\alpha$3 Bus in EUR | $\alpha$4 Walking in EUR | $\alpha$5 Scooter in EUR | $\beta$ in EUR/s |
|---|---|---|---|---|---|
| 0 | −0.2604 | 1.855 | 2.016 | −0.0761 | 0.002 |

The simulated traffic flows were validated against the actual detector flows distributed across the city, as shown in Figure 5a. The 592 detectors that counted the average morning

peak hour flow on a working day in October 2022 between 7:00 a.m. and 8:00 a.m. [66] were used for the validation process. It is noted that the detectors that could not be matched to road links were excluded from the validation process to avoid comparisons with incorrect link flows.

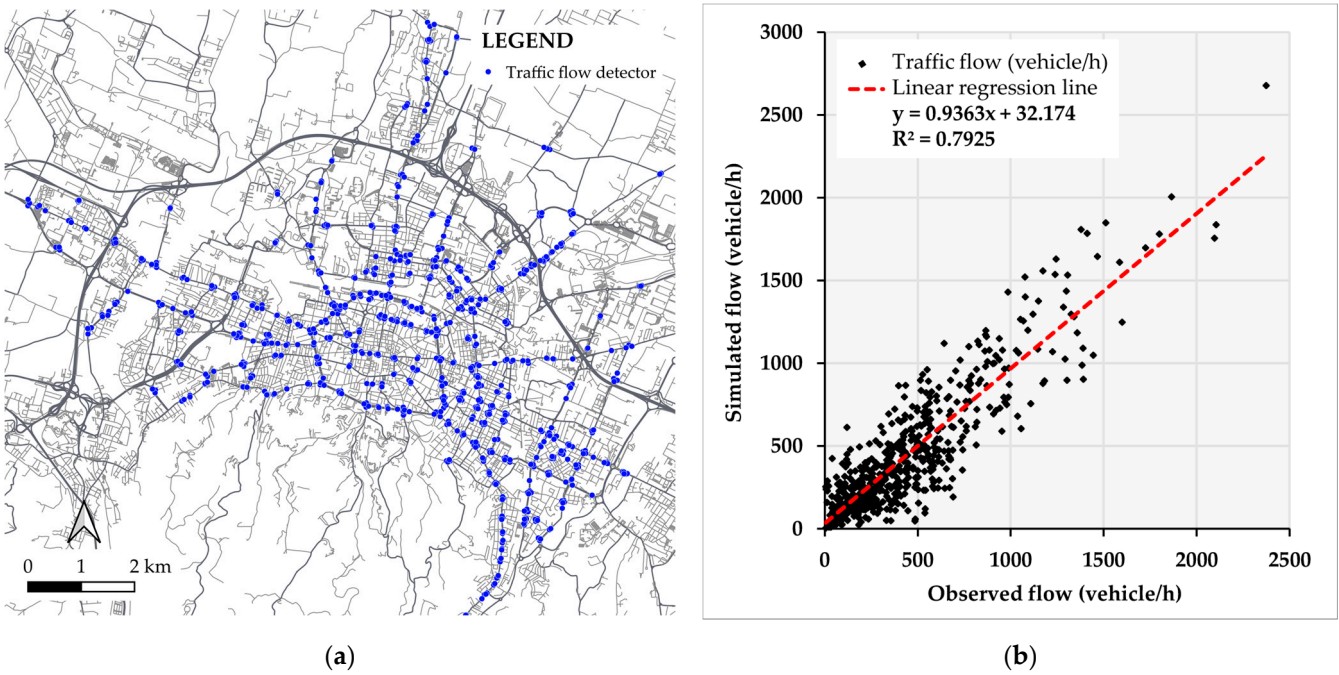

(**a**)　　　　　　　　　　　　　　　　　　　(**b**)

**Figure 5.** (**a**) Representation of the distribution of 592 flow measurement detectors in Bologna (blue dots) used to calibrate the microscopic traffic model. (**b**) The regression diagram shows the simulated traffic flow compared to the observed flow (vehicle/h).

The validation procedure is based on three indicators defined in [15]. Figure 5b shows the regression analysis between simulated and observed flows, which indicates a slope of m = 0.9363 and an intercept of 32 vehicles per hour. The two parameters are significant, with *p*-values of $3.999 \times 10^{-3}$ for the slope and $1.235 \times 10^{-203}$ for the intercept. The simulated slope in the range of 0.9 < m < 1.1 is acceptable, and the coefficient $R^2$ = 0.7925 is approximately at the acceptance level of $R^2$ > 0.8, according to [55], showing a relatively strong agreement between the simulated and the observed traffic flows. Finally, the GEH statistic was also retested to measure the absolute and percentage differences between the modeled and observed flows using thresholds explained in [55]. The results show an improvement in the goodness-of-fit of the updated model compared to the one in [15], as the rates of links below 5, from 5 to 10, and 10 and above are 44.59%, 22.30%, and 33.11%, respectively.

### 2.5. Implementing Bologna Superblocks Corresponding to Traffic Intervention Measures

The superblocks proposed in Section 2.3 represent an area for the implementation of traffic intervention measures. A group of five measures were introduced, aiming at reducing motorized traffic circulation, facilitating active transport, and enhancing travel experience for PT users, including (i) prohibiting private car access and reducing speed limits on local roads; (ii) eliminating on-street parking on local roads; (iii) providing centralized parking on main roads; (iv) implementing more zebra crossings, improving pedestrian and bicycle paths within the superblocks; and (v) reorganizing bus routes and increasing service frequency.

Algorithms were developed to filter all road lane objects and associated attributes within the 49 superblocks, then remove car access (9044 lane objects), and apply the speed limit of 10 km/h (4229 lane objects), as well as remove 15,700 on-street parking spots

within the superblocks. In addition, three criteria were established to identify feasible locations for centralized parking substitution for the on-street parking removal, focusing on (i) user accessibility—locating near the superblock boundaries and bus stops to encourage intermodal transfers; (ii) land availability—locating on existing private parking to utilize existing capacity or correspond to large vacant areas; and (iii) the city's parking network plan—aligning with planned centralized parking areas. Using satellite images from Google Maps, the city's existing parking data [59], and on-site inspections, 43 centralized parking lots with a total capacity of 15,700 lots were identified for modeling.

In addition, to prioritize active mobility, pedestrian crossings, footpaths, and cycle paths within the superblock-affected area were enhanced. All bus routes crossing the superblocks were deviated to main roads, and associated bus stops were also relocated where necessary to guarantee a spacing of 400 m. Lastly, the intervals were reduced to under 5 min, achieving a comfortable waiting time of 2–3 min. As a result, 20 city bus lines were partially reorganized, and for 40 lines, the headway was reduced.

All the changes to the supply model were implemented, and the demand was added and successfully tested. Ten micro-simulation iterations were carried out to allow people to experience all feasible plans. Finally, the calibrated mode-specific parameters of the utility function from Table 2 were applied to determine the plan choices of the synthetic population of the superblock scenario.

## 3. Simulation Results

This section presents the results of the microscopic traffic simulation for the baseline and superblock scenarios. The changes in mobility at network and trip levels are estimated by comparing indicators before and after the introduction of the superblocks (see Table 3). It is worth mentioning that these are short- and medium-term changes, as people are assumed to maintain their current activity locations. The network-level mobility benefit analysis focuses on estimating the modal shift between private vehicles (i.e., cars, motorcycles) and green transport modes (i.e., public buses, bicycles, and walking), in addition to changes in traffic performance (i.e., traffic volume, traffic density, and travel speed). Meanwhile, the analysis of trip-level benefits focuses on the changes in door-to-door travel time for a single person to/from the superblocks. Lastly, the improvements in traffic-related air emissions are also estimated.

**Table 3.** Targeted areas for evaluating the mobility benefits of the superblocks and the proposed evaluation indicators.

| # | Targeted Areas | Area (km²) | Population | Number of Edges | Mobility Evaluation Indicators | | | |
|---|---|---|---|---|---|---|---|---|
| | | | | | Modal Share | Door-to-Door Travel Time | Traffic Performance | Traffic-Related Air Emissions |
| 1 | Citywide | 50.0 | 167,062 | 32,409 | ☑ | ☐ | ☐ | ☐ |
| 2 | Affected area | 23.8 | 121,509 | 9480 | ☑ | ☐ | ☑ | ☑ |
| - | Within superblock | 10.7 | 69,950 | 4711 | ☐ | ☑ | ☑ | ☑ |
| - | Adjacent area | 13.1 | 51,559 | 4769 | ☐ | ☐ | ☑ | ☑ |

Note: Population refers to the number of people who have either their origin or destination or both located within the targeted area. The indicators used to evaluate the mobility benefits of the superblock model in the targeted area: ☑—applicable, ☐—not applicable.

### 3.1. Evaluations of Modal Shifts at the Citywide and Superblock Levels

Our analysis of modal shifts at the citywide level (N = 167,062 people) highlighted a significant change in the modal share of cars shifted to buses after the introduction of the 49 superblocks, as shown in Figure 6a. The car usage in the baseline scenario is about 35.0%, while this dropped significantly by 5.4% (significant, $p < 0.05$) to 29.6% in the superblock scenario. Interestingly, the share of bus usage increased significantly by 5.7% ($p < 0.05$) from 25.5% to 31.3%. The modeling results also indicated that the modal shares of non-motorized transport modes, including cycling and walking, increased slightly, by 0.4% ($p < 0.05$), from

7.0% to 7.4%, and by 0.8% ($p < 0.05$), from 22.8% to 23.6%, respectively. In contrast, the number of motorcyclists fell by 1.6%, from 10.6% to 9.0%.

**Figure 6.** Changes in modal shares at the citywide (**a**) and superblock (**b**) levels before and after the introduction of superblocks.

The changes in the mode shares in the superblock area (N = 69,950), equivalent to 41.9% of the citywide population, were also investigated, as shown in Figure 6b. After the introduction of the superblocks with significant improvements in the service frequency, the bus share within the superblock area increased by 6.5% ($p < 0.05$), to 39.2%. Consequently, the shares of private vehicles (i.e., cars and motorcycles) decreased significantly by 4.6% ($p < 0.05$), from 22.2% to 17.6%, and by 2.2% ($p < 0.05$), from 10.7% to 8.5%, respectively. However, the differences in the shares of non-motorized modes (i.e., bicycles and walking) are not significant, as the bicycle trips remained unchanged at 6.9%, and the walking trips increased by only 0.3% (not significant, $p > 0.05$).

The cumulative distribution function (CDF) of the total travel time by mode was analyzed before and after the introduction of the superblocks at citywide and superblock levels. It was observed that bus users have the longest travel time, followed by cyclists, car drivers, motorcyclists, and pedestrians, regardless of modes and analyzed areas.

At the citywide level, in the baseline scenario, about 80% and 50% of bus users have a travel time of more than 20 and 30 min/trip, respectively. Meanwhile, only about 55% and 22% of cyclists, 40% and 9% of car drivers, and less than 15% and 5% of motorcyclists travel longer than these thresholds. The travel time of bus users was reduced in the superblock scenario, with the percentage of bus trips longer than 20 and 30 min being reduced to below 73% and 40% compared to the baseline travel time. Figure 7a,b show in detail the CDF of total travel time at the superblock level. The share of bus trips taking longer than 20 and 30 min dropped from 78% and 40% in the baseline scenario to about 70% and 37% in the superblock scenario. In addition, the travel time of car drivers increased overall as about 40% and 9% of car trips traveled over 20 and 30 min compared to only 30% and 5% of car trips in the baseline scenario.

However, the changes in the CDF of trip-level patterns for bus users and car drivers, as well as the total travel times for cyclists, motorcyclists, and pedestrians at both the citywide and superblock levels, are not readily apparent and are further investigated and discussed in the trip-level analysis in Section 3.2.

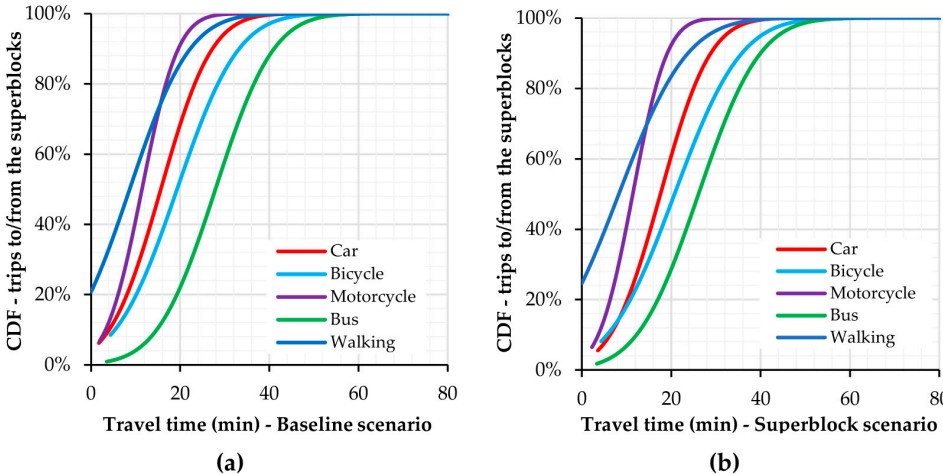

**Figure 7.** Cumulative distribution function (CDF) of the door-to-door travel time by modes of the trips generated/terminated within the superblock area before (**a**) and after (**b**) the introduction of superblocks.

### 3.2. Evaluations of Door-to-Door Travel Time

A more thorough examination of the door-to-door travel times at trip level for various travel groups was conducted by analyzing simulation results for individuals who completed their trip between 6:00 a.m. and 8:00 a.m. in the baseline (N = 39,676 people) and superblock (N = 39,159 people) scenarios. This investigation revealed significant changes in both total travel time and different stages of the journey—defined in Figure 2.

Table 4 shows the mean, median, and statistically significant values of door-to-door travel time by mode for all the trips to/from the superblock area in the two scenarios, while Figure 8 depicts the interquartile ranges (IQRs) of the trip modes, which are significantly different in door-to-door travel time. The average door-to-door travel time of a single person, regardless of mode usage, in the superblock scenario increased by 8.93% (mean: 15.08, IQR: 4.42–21.40 to mean: 16.42, IQR: 4.42–24.78 min/trip).

**Table 4.** Number of simulated trips by modes to and from the superblocks from 6:00 a.m. to 8:00 a.m. and the door-to-door travel time per trip in the superblock scenario (values in **bold**), compared to those of the baseline scenario (values in parentheses).

| # | Door-to-Door Travel Time per Trip to/from the Superblocks by Modes | | Number of Trips | Mean (min/trip) | Median (min/trip) | Std. Deviation | T-Test |
|---|---|---|---|---|---|---|---|
| 1 | All mode | Total travel time | **39,159** (39,676) | **16.43** (15.08) | **13.93** (12.03) | **13.73** (12.03) | *** |
| 2 | Car trip | Total travel time | **5785** (8637) | **20.59** (16.65) | **18.6** (14.43) | **9.67** (8.98) | *** |
| | | Walking time | | **5.42** (2.95) | **3.85** (1.73) | **5.36** (4.18) | *** |
| | | Riding time | | **14.31** (13.15) | **12.68** (11.28) | **7.55** (7.55) | n.s. |
| 3 | Bus trip | Total travel time | **10,253** (6548) | **29.98** (32.78) | **28.73** (30.85) | **11.95** (13.72) | *** |
| | | Walking time | | **7.12** (6.85) | **5.86** (5.57) | **5.44** (5.10) | *** |
| | | Riding time | | **16.27** (16.6) | **15.23** (15.60) | **8.49** (8.77) | ** |
| | | Waiting time | | **6.86** (9.06) | **6.18** (7.82) | **4.84** (6.88) | * |
| 4 | Bicycle trip | Total travel time | **2257** (2545) | **24.03** (23.48) | **21.2** (20.65) | **11.78** (11.48) | *** |
| 5 | Motorcycle trip | Total travel time | **4102** (4943) | **11.98** (11.74) | **10.28** (10.18) | **6.7** (6.35) | n.s. |
| 6 | Walking | Total travel time | **16,739** (17,003) | **6.77** (7.18) | **2.83** (2.83) | **8.16** (8.86) | ** |

Note: Statistically significant levels for travel time in the superblock scenario compared to the baseline scenario: 0, '***'; 0.001, '**'; 0.005, '*'; not significant, 'n.s.'

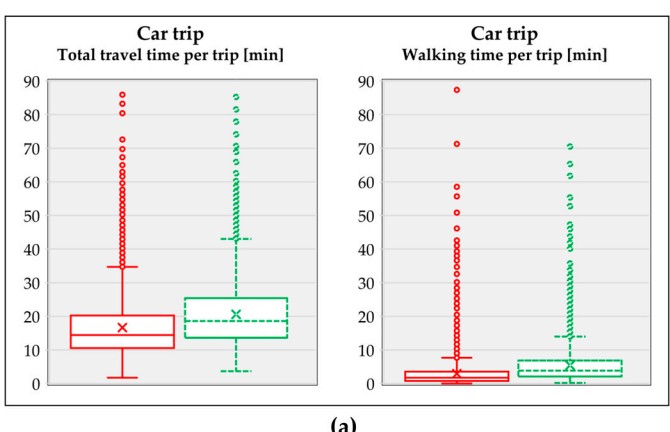
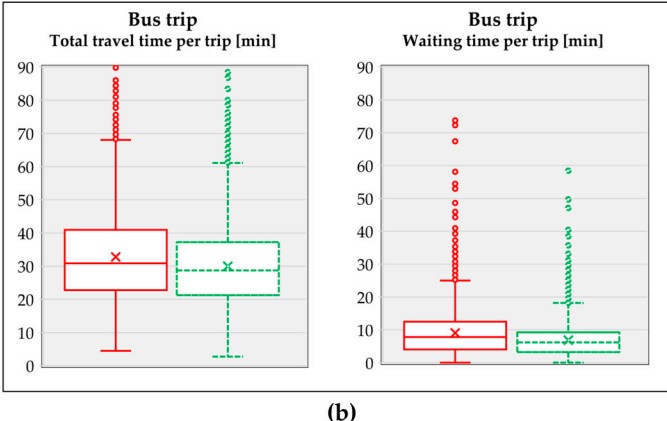

**Figure 8.** Significant changes in mean and IQR of the door-to-door travel time to and from the superblocks of car users (**a**) and bus users (**b**) in the superblock scenario (green box) compared to that of the baseline scenario (red box).

In terms of differences in total travel time by mode, it is noteworthy that the differences in total travel time before and after the introduction of the superblocks between the different groups were significant ($p < 0.05$), except for motorcyclists. The analysis showed that the car traffic restriction measures not only forced car users to shift to buses (see Section 3.1) but also increased their average overall travel time by 23.66% (mean: 16.65, IQR: 10.57–20.23 to mean: 20.59, IQR: 13.64–25.42 min/trip). The change was particularly marked in the average walking time of car users, with an increase of about 83.73% (mean: 2.95, IQR: 0.78–3.54 to mean: 5.42, IQR 2.10–6.87 min/trip), while the changes in riding time are not significant ($p > 0.05$).

In terms of green transport modes, the increased frequency of bus service had significant impacts on the door-to-door travel time of bus users. The average total travel time of a bus trip dropped by about 9.34% (mean: 32.78, IQR: 22.78–40.96 to mean: 29.98, IQR: 21.25–37.23 min/trip), and the average waiting time at bus stops decreased by 32.07% (mean: 9.06, IQR: 4.02–12.50 to mean: 6.86, IQR: 3.25–9.23 min/trip). The simulation results also revealed that pedestrians benefited from the improvements to pedestrian facilities as their travel time was reduced by about 6.06% (mean: 7.18. IQR: 1.18–10.77 to mean: 6.77, IQR: 1.22–10.13 min/trip). In addition, the results showed a slight increase in average riding times, from 23.48 to 24.03 min per trip for cyclists and 11.74 to 11.98 min per trip for motorcyclists.

### 3.3. Evaluations of Traffic Performance and Traffic-Related Air Emissions

A performance analysis of the road network within the affected area in one morning-peak hour (from 7:00 a.m. to 8:00 a.m.) was conducted as indicated in Table 5. It concerns 4711 road links within the superblock boundaries as well as 4769 adjacent roads. The analysis revealed a 13.94% decrease in the number of vehicles entering the affected road links and an 8.09% decrease in average traffic densities after the implementation of the 49 superblocks. There was a significant reduction of 48.48% in traffic volume and 49.95% in traffic density within the superblock area. Surprisingly, there was also a 6.09% decrease in flows on the adjacent roads. The average travel speed in the affected areas appears to have decreased, but not significantly.

Figure 9 illustrates the spatial distribution of vehicles per road link between 7:00 a.m. and 8:00 a.m. in the baseline and superblock scenarios. In the baseline scenario (Figure 9a), the traffic flows in the superblock area (highlighted in the light pink background) are generally below 1500 vehicles/h per road link (shown in blue and gray), except for some major eastern and western trunk routes. Higher traffic volumes, exceeding 2500 vehicles/h, are seen on the outer ring road, or "Tangenziale", represented by the red color. The reduction in the number of traffic flows in the superblock scenario (Figure 9b) is shown by

the decrease in the density of the gray lines within the superblock area. The diversion of external traffic is indicated by an increase in the density of red lines on the "Tangenziale".

**Table 5.** Changes in traffic volume, traffic density, and travel speed in the superblock scenario (values in **bold**), compared to the baseline scenario (values in parentheses).

| # | Targeted Areas | Abs. Entered Flows ($10^3$ veh./h) | Changes in Entered Flows (%) | Av. Traffic Density (veh./km) | Changes in Traffic Density (%) | Av. Travel Speeds (km/h) | Changes in Travel Speed (%) |
|---|---|---|---|---|---|---|---|
| 1 | Affected Area | **1270** (1475) | −13.94% | **245** (267) | −8.09% | **22.17** (22.20) | −0.14% |
| 2 | Within superblock | **141** (273) | −48.48% | **46** (92) | −49.95% | **21.46** (21.48) | −0.09% |
| 3 | Adjacent roads | **1129** (1202) | −6.09% | **432** (435) | −0.78% | **22.79** (22.87) | −0.35% |

Note: Abs., absolute; Av., average; veh., vehicle.

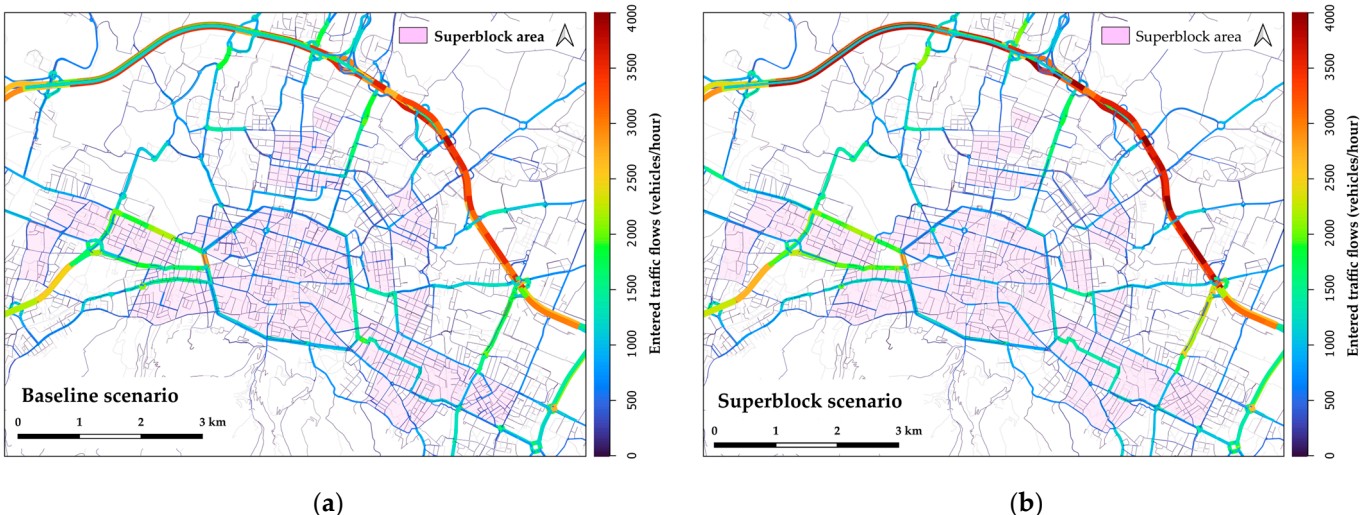

**Figure 9.** Simulated entered traffic link flows in the affected area as the number of traffic vehicles that entered into a road link from 7:00 a.m. to 8:00 a.m. in the baseline (**a**) and superblock (**b**) scenarios.

In the microscopic traffic model, traffic-related air emissions were calculated based on the pollutant emission model in SUMO reported in [62], which uses data from the Handbook Emission Factors for Road Transport (HBEFA) to calculate emissions based on vehicle type, speed, and slope of the road during the simulation, allowing us to analyze the environmental impacts of different traffic intervention scenarios on specific road links over time. The key air emission indicators include fuel consumption, CO, $CO_2$, and PMx. Absolute emissions were estimated based on the link flow emission results from 7:00 a.m. to 8:00 a.m. The absolute emissions of every link flow and aggregate value before and after the introduction of the superblock were compared. The results indicate a significant decrease in absolute air emissions for all indicators within the 49 superblocks, including fuel consumption (68.77%), CO (64.72%), $CO_2$ (68.32%), and PMx (79.01%). Consequently, the air emission indicators in the entire affected areas decreased accordingly, ranging from 11.13% to 18.97%. Interestingly, the emission levels on adjacent roads also showed improvements, decreasing in a range from 2.15% to 8.17%, except for PMx.

## 4. Discussion

The multiple-criteria approach used to identify 49 feasible superblocks in Bologna city could be applicable to any city, as demonstrated in [33,37,38,49]. However, the configuration of potential superblocks may vary depending on the specific urban design.

The improvement in goodness-of-fit of the updated microscopic traffic simulation model with respect to our previous model [15] and the calibrated plan choice model

allowed us to generate the baseline mode shares aligning to the latest city mode shares as reported in [57]. The baseline mode shares of active mobility and bus services in the superblock area are significantly higher than that at the citywide level, possibly because the superblocks are in the urban core of Bologna, well-equipped with footpaths and cycle paths, and where the bus network has better coverage with respect to the periphery and offers more frequent service. These fundamental factors facilitated the implementation of the superblock scenario, the modeling of all the traffic intervention measures, and the identification of mobility impacts of the proposed 49 superblocks.

The simulation results indicate that the introduction of superblocks has both global and local impacts on the mode-choice behaviors of users. The trips made by private vehicles are gradually replaced by sustainable transport modes. This result is meaningful in that the traffic restriction measures and the relocation of on-street parking spaces result in drivers having to walk to the centralized parking (83.73% increase in walking time) and deviating from their original routes to their destination. Consequently, the travel time by car does increase by 23.66%. Meanwhile, the increased bus frequencies make the bus service more attractive, as bus users experience shorter waiting times at the bus stops (32.07% decrease in waiting time), resulting in a 9.34% decrease in their door-to-door travel time. Obviously, our finding of a reduction in car trips is conservative relative to the Barcelona studies in [9,11,34,43].

The shares of bicycles, motorcycles, and walking in the superblocks did not change significantly, although the results showed a slight increase in average riding times for cyclists and motorcyclists. This could be due to the low-speed limit of 10 km/h within the superblocks—these may have slowed down their average travel speed and increased travel times. In short, the improvements in active mobility infrastructure and associated car traffic restriction measures are not part of the mode-share model; thus, walking and cycling cannot profit from longer door-to-door times of car trips. However, the increased perception of road safety may increase cycling and walking, but this is subjective behavior and not part of the present model.

The expected reductions in traffic volume and traffic density within the superblocks and on the affected road links can be attributed to the citywide and superblock-level car-to-bus mode shift, as discussed in Section 3.1. In addition, traffic restriction measures were implemented within the superblocks, which disallow cross-through traffic within superblock boundaries. Car and motorcycle users must walk (non-motorized traffic) from the centralized parking to their destination, which could be a reason for the reduction in traffic flows on the main roads. Moreover, to avoid traffic jams, external vehicles from the periphery are ready to accept larger deviations instead of crossing through the city center. The present micro-simulation approach has thus addressed the questions raised in [11] of whether superblocks increase traffic congestion and create new bottlenecks on the main roads. In addition, these findings imply that cities surrounded by ring roads would be more ideal for implementing the superblocks to avoid traffic congestion on main roads and at intersections caused by cross-through traffic.

Lastly, the environmental benefits of the superblocks could be explained by the sustainable model shift, the lower traffic flows in superblock areas, and the shorter riding time of motorized traffic after the introduction of the 49 superblocks.

In practice, these results provide scientists and urban and transport planners with more insights into how the changes in door-to-door travel times of multi-modal trips can impact individual travel behavior and traffic performance at a citywide level. The mobility benefits of the superblock model have been demonstrated: sustainable transport modes are facilitated without generating negative traffic-related impacts in adjacent areas.

## 5. Conclusions

The present research addressed the current research gap in modeling the door-to-door travel time of multi-modal trips and evaluating the mobility benefits of the superblock in

larger urban areas. Different measures of the integrated "avoid", "shift", and "improve" strategy were modeled and evaluated.

In particular, changes in door-to-door travel times for multi-modal trips and how these changes affect individual travel behavior and mode choice before and after the introduction of superblocks were assessed by using a microscopic simulation approach.

The superblocks themselves were identified by applying a multiple-criteria approach, which could be applicable to any city. In the case of Bologna city, 49 feasible superblocks were identified. A baseline scenario representing the business-as-usual situation was built, improved, and validated to capture the city's latest mode shares and estimate the door-to-door travel time of different vehicle users. A traffic intervention scenario, called a superblock scenario, was created to model a group of five measures, which is focused on reducing motorized traffic circulation, promoting active mobility, and enhancing the travel experience for PT users within the 49 superblocks. Then, the mobility benefits were estimated by comparing indicators before and after the introduction of the superblocks, focusing on the changes in modal shift, door-to-door travel time, traffic performance, and improvements in traffic-related air emissions at both citywide and trip levels.

The study found significant impacts of the traffic intervention measures within the 49 superblocks on the total door-to-door travel time of car and bus users. The traffic restrictions within the superblocks resulted in a significant increase of 83.73% in walking time for car drivers, while car-riding time remained relatively unchanged. As a result, the total travel time for car drivers increased by 23.66%. On the other hand, bus users benefited from the increased frequency of bus services on main roads, leading to a decrease in average waiting time by 32.07%. This resulted in a decrease of 9.34% in their door-to-door travel time. These changes have a significant impact on the mode choice with a noticeable shift from cars to buses. The citywide and superblock mode-share analysis showed reductions in car trips by 5.4% and 4.6%, respectively. This led to an increase in bus trips by 5.7% (and 6.5% in the superblock area) after the implementation of superblocks. The shares of bicycles and walking did not change significantly; it seems that walking and cycling cannot profit from the longer door-to-door times of car trips, but cycling may increase due to a safer environment.

The research also showed that absolute traffic volumes and traffic-related emissions in the superblock-affected areas decreased significantly. Surprisingly, traffic volumes on the roads around the superblocks did not increase as expected; instead, they decreased slightly by 6.09%. These positive benefits are mainly due to the car-to-bus shift, while the car driving time did not change significantly and there were large deviations in external traffic from the city center. This would imply that cities surrounded by ring roads would be more ideal for implementing the superblocks.

Therefore, the superblock model could be considered as one feasible and effective measure for achieving net-zero carbon by 2050, as committed to in the 2015 Paris Agreement. However, this study still has limitations in modeling the long-term effects of the superblocks, such as changes in the activity locations of the people who have either their origin or/and destination located within and between the superblocks.

**Author Contributions:** Conceptualization, Ngoc An Nguyen and Joerg Schweizer; methodology, Ngoc An Nguyen and Joerg Schweizer; software, Joerg Schweizer and Ngoc An Nguyen; validation, Ngoc An Nguyen and Joerg Schweizer; formal analysis, Ngoc An Nguyen; investigation, Ngoc An Nguyen and Joerg Schweizer; resources, Joerg Schweizer and Ngoc An Nguyen; data curation, Ngoc An Nguyen, Sofia Palese and Leonardo Posati; writing—original draft preparation, Ngoc An Nguyen, Joerg Schweizer, Sofia Palese and Leonardo Posati; writing—review and editing, Ngoc An Nguyen, Joerg Schweizer and Federico Rupi; visualization, Ngoc An Nguyen; supervision, Ngoc-An Ngu-yen and Joerg Schweizer; project administration, Ngoc An Nguyen; funding acquisition, Joerg Schweizer and Federico Rupi. All authors have read and agreed to the published version of the manuscript.

**Funding:** This research was funded by the ECOSISTER project, Spoke 4, and the Italian PNRR program.

**Data Availability Statement:** The data that support the findings of this study are available on request from the corresponding author.

**Conflicts of Interest:** The authors declare no conflicts of interest.

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
