# Peer review of "Superblock Design and Evaluation by a Microscopic Door-to-Door Simulation Approach"

_ijgi, doi:10.3390/ijgi13030077_

Round 1

Reviewer 1 Report

Comments and Suggestions for Authors

This is an interesting and insightful work that developing a microscopic door-to-door simulation approach for superblock design and evaluation. However, there are still some shortcomings in this manuscript.

1. Please point out the research gap in modeling superblocks in the first part of Abstract section.

2. Suggest to shortening the Introduction section within three pages.

3. the Method section is also too long.

4. figure 6, suggest to move (a) and (b) to the upper left corner of each sub-figure.

5. Suggest to separate the Result and Discussion section.

6. Please make the conclusion more concise and deepen.

Comments on the Quality of English Language

It is easy to understand.

Author Response

Dear International Journal of Geo-Information Reviewers

Thank you for giving us the opportunity to submit a revised draft of my manuscript titled “Superblock design and evaluation by a Microscopic Door-to-Door Simulation Approach” to the International Journal of Geo-Information. We appreciate the time and effort that you and the reviewers have dedicated to providing your valuable feedback on our manuscript. We are grateful to the reviewers for their insightful comments on the paper. We have been able to incorporate changes to reflect most of the suggestions provided by the reviewers.

Here is a point-by-point response to the reviewers’ comments and concerns.

  1. Comments from Reviewer 1
  • General comment: This is an interesting and insightful work that developing a microscopic door-to-door simulation approach for superblock design and evaluation. However, there are still some shortcomings in this manuscript.
  • Comment 1: Please point out the research gap in modeling superblocks in the first part of Abstract section.

Response: We thank the Reviewer for this suggestion: the research gap has been stated clearer and updated in the Abstract section as follows: “ The present study contributes to narrow down the research gap in modeling individual door-to-door trips in a superblock scenario and in evaluating the respective impacts in terms of travel-times, modal shifts, traffic performance and environmental benefits. The methods used are a multiple-criteria approach to identify the superblocks and a large-scale, multi-model, activity based microscopic simulation.”

  • Comment 2: Suggest to shortening the Introduction section within three pages.

Response:  We agree with the Reviewer, now the “Introduction” section consists of 2 sub-sections including “Introduction to Superblocks and Their Analysis” and “State of the Art”, we have refined and shortened to almost 3 pages of total full 17-page paper (without reference section), so the reader can grasp the contents faster.

  • Comment 3: The Method section is also too long.

Response: We thank for this suggestion: given the broad study content, the “Materials and Methods” divided into 5 sub-sections including “Microscopic Door-to-Door Simulation Approach”, “Superblock Configuration Identification”, “Bologna Study Area and Proposed Superblock Configuration”, “The Baseline Scenario and Its Validation”, and “Modelling Bologna Superblocks Corresponding to Traffic Intervention Measures”. We have refined and shortened into almost 6 pages of total full 17-page paper (without reference section).

  • Comment 4: Figure 6, suggest to move (a) and (b) to the upper left corner of each sub-figure.

Response: To respect the template, the figure numbers placed under the corresponding figures, which are consistent to the remaining figures following the template.

  • Comment 5: Suggest to separate the Result and Discussion section.

Response:  We agree with the Reviewer, the “Results and Discussion” section has been separated as indicated in the “Section 3 – Simulation Results” and “Section 4 – Discussion”

  • Comment 6: Please make the conclusion more concise and deepen.

Response: We thank for this suggestion: the conclusion has been refined to more concise and deeper as indicated in “Section 5 – Conclusion”

  • Comment 7: Comments on the Quality of English Language, It is easy to understand.

Response:  We would like to thank the Reviewer for such constructive comments to improve the overall quality of this manuscript.

Reviewer 2 Report

Comments and Suggestions for Authors

In this article, the authors used a simulation model to evaluate the effects of superblocks on traffic patterns and traffic users behavior within the city areas. It seems an interesting topic that has the value for consideration. The paper is well written. Although this study proposed a well understandable framework for evaluating different scenarios for road traffic improvement within residential network area, however, more complementary research hypotheses may need to be studied for more accurate analysis and assessment. There are some points that I feel they can improve the current study:

- Literature can be improved by adding more related references from peer-reveiwed journals.

- Other mode choices such as subway or airport can also be considered in the model. Various area amount of superblocks may also have different impact on the model performance that can be considered in the future studies. Selecting proper parking positioning criteria may need a separate study as the model input.

- Table 3 is missing.

- Real implementation of the superblock scenario may improve the validity of the obtained traffic attribute changes at baseline and superblock scenarios.

- The variation of air emission indicators may need to be explained more on how they were calculated and compared.

Comments on the Quality of English Language

- Lines 65 to 68 needs to be rewrite for better understanding.

- In line 619, “by using” were repeated twice.

Author Response

Dear International Journal of Geo-Information Reviewers

Thank you for giving us the opportunity to submit a revised draft of my manuscript titled “Superblock design and evaluation by a Microscopic Door-to-Door Simulation Approach” to the International Journal of Geo-Information. We appreciate the time and effort that you and the reviewers have dedicated to providing your valuable feedback on our manuscript. We are grateful to the reviewers for their insightful comments on the paper. We have been able to incorporate changes to reflect most of the suggestions provided by the reviewers.

Here is a point-by-point response to the reviewers’ comments and concerns.

  1. Comments from Reviewer 2
  • General comment: In this article, the authors used a simulation model to evaluate the effects of superblocks on traffic patterns and traffic users behavior within the city areas. It seems an interesting topic that has the value for consideration. The paper is well written. Although this study proposed a well understandable framework for evaluating different scenarios for road traffic improvement within residential network area, however, more complementary research hypotheses may need to be studied for more accurate analysis and assessment. There are some points that I feel they can improve the current study.
  • Comment 1: Literature can be improved by adding more related references from peer-reveiwed journals.

Response: We thank the Reviewer for this suggestion: in the Literature works, we reviewed all the relevant and most up-to-date studies of the superblock models including peer-reviewed journals. The literature review is shown in Section 1.2 – State of the Art.

  • Comment 2: Other mode choices such as subway or airport can also be considered in the model. Various area amount of superblocks may also have different impact on the model performance that can be considered in the future studies. Selecting proper parking positioning criteria may need a separate study as the model input.

Response: We thank the Reviewer for raising this concern. In our study we focused on the use of the public road space and limited the travel purpose to home-work trips, because this demand is covered by the OD matrices. Even though we do have the OD matrices for rail, we decided that the modelling of the rail sub-system is not worth the effort, as rail passengers do not significantly interfere with road users (they do interfere but only in the form of pedestrians).  More specifically, Bologna’s regional train stations, serving external travel demand, is located at the northern edge of the superblock area. Again, these train users mainly utilize public buses or walking for first/last-mile connections. Thus, they are not involved in the motorized traffic restriction measures within the superblocks and do not interfere with motorized road traffic.

Regarding the demand induced by the airport: due to its suburban location outside the city’s outer ring road and superblock coverage, the Bologna airport primarily serves tourists – who rely on public buses getting to the city. Also, this is a non-systematic demand for which we have no data. But like the rail passengers, airport-users would use the monorail to get into the center and would not interfere with the road space of the superblocks in the city core.  

 However, we certainly agree that the impacts of superblocks and centralized parking position on the modal split (including external rail passengers) need to be investigated in detail for further studies. In fact, the next step of the microsimulation model is to include the rail network and rail schedules of all local trains.

  • Comment 3: Table 3 is missing.

Response: We thank the Reviewer for this comment: Table 3 was missed in the first draft, all the tables have been updated with the number following the order.

  • Comment 4: Real implementation of the superblock scenario may improve the validity of the obtained traffic attribute changes at baseline and superblock scenarios.

Response: This is our first study on superblock model in Bologna, Italy. The city is introducing low-emission zones and planning speed limits in urban core to promote active mobility. A car-restricted area (for non-residents), called “green shield” is also envisaged in the superblock area, which resembles the superblocks, but with slightly different rules.  With the developed method we can predict the outcome of the superblock scheme before it gets implemented, these are useful results which the city plan may take into account. It will certainly be interestinfg to validate the prediction with  real world data, but unfortunately, we are not in the position to do it at this time.

  • Comment 5: The variation of air emission indicators may need to be explained more on how they were calculated and compared.

Response: We thank the Reviewer for this comment: the description of the air emission model used in the study has been added (line 539 to line 546). The used SUMO emission model is based on the Handbook Emission Factors for Road Transport (HBEFA) to calculate emissions in function of vehicle type, speed and slope of the road during the simulation.

  • Comment 6: Comments on the Quality of English Language: Lines 65 to 68 needs to be rewrite for better understanding; In line 619, “by using” were repeated twice.

Response: The Reviewer is right: sentence in the line 61 to 63 has been refined as follows for easier understand: “The models would be able to adequately represent the changes in the transport supply in a what-if scenario, as well as simulate and quantify its impacts”

The typo in the line 619 has been refined and moved into the Section 5 – Discussion, line 595 to line 597

We would like to thank the Reviewer for such constructive comments to improve the overall quality of this manuscript.

Reviewer 3 Report

Comments and Suggestions for Authors

Congratulations to the authors. The results of this work are very useful for mobility planning in large cities.

Author Response

Dear International Journal of Geo-Information Reviewers

Thank you for giving us the opportunity to submit a revised draft of my manuscript titled “Superblock design and evaluation by a Microscopic Door-to-Door Simulation Approach” to the International Journal of Geo-Information. We appreciate the time and effort that you and the reviewers have dedicated to providing your valuable feedback on our manuscript. We are grateful to the reviewers for their insightful comments on the paper. We have been able to incorporate changes to reflect most of the suggestions provided by the reviewers.

Here is a point-by-point response to the reviewers’ comments and concerns.

  1. Comments from Reviewer 3
  • General comment: Congratulations to the authors. The results of this work are very useful for mobility planning in large cities.

Response: We really appreciate your comment.